# Family Physicians Working at Hospitals: A 20-Year Nationwide Trend Analysis in Taiwan

**DOI:** 10.3390/ijerph18179097

**Published:** 2021-08-28

**Authors:** Yueh-Hsin Wang, Hui-Chun Li, Kuang-Yu Liao, Tzeng-Ji Chen, Shinn-Jang Hwang

**Affiliations:** 1Department of Family Medicine, Taipei Veterans General Hospital, Taipei 112, Taiwan; yhsin.wang@gmail.com (Y.-H.W.); huinchin@gmail.com (H.-C.L.); yu203209@gmail.com (K.-Y.L.); sjhwang@vghtpe.gov.tw (S.-J.H.); 2School of Medicine, National Yang Ming Chiao Tung University, Taipei 112, Taiwan

**Keywords:** family physicians, professional practice location, supply and distribution, Taiwan

## Abstract

Family physicians play an essential role as gatekeepers in primary health care. However, most studies in the past focused on the geographic maldistribution of family physicians, and few studies focused on the distribution of family physicians between private practices and hospitals. This study aims to analyze the trends in practice locations of family physicians in Taiwan between 1999 and 2018, using the databases of the Taiwan Association of Family Medicine and Taiwan Medical Association. Although the annual number of physicians registered as family physicians had steadily increased from 1876 in 1999 to 3655 in 2018, the ratio of family physicians practicing in hospitals to total family physicians remained stable around 40% in the study period. Even after eliminating the trainees who were entirely registered at hospitals, the proportion of hospital-based family physicians still accounted for about one-third of the total in each year. In conclusion, family physicians had been continuously demanded by hospitals in Taiwan. If the supply of primary care-oriented family physicians is insufficient outside hospitals, health manpower planning would require urgent adjustments.

## 1. Introduction

Family physicians, as primary care physicians, are the gatekeepers of patients’ health [1,2]. They provide comprehensive, continuous, coordinated, accessible, and accountable care [3]. Family physicians provide a wide range of holistic medical care, including acute and chronic treatment, hospice and palliative care, and preventive medicine [4]. In Western countries, most family physicians work at physician clinics instead of hospitals [5,6]. A minority of family physicians work at community or remote hospitals, where family physicians may be among the few physicians employed in these sparsely populated areas. However, with advances in health care, a gradual shift toward specialist care in community hospitals, and an increased demand for trans-disciplinary care, the proportion of family physicians working at hospitals is gradually decreasing [7,8]. In Canada, for example, the Canadian Medical Association Physician Workforce Survey reported a drop from 11.6% to 9.2% in family physicians serving in community hospitals between 2017 and 2019 [9]. Most studies in the past focused on the geographic maldistribution of family physicians, and few focused on the distribution of family physicians between private practices and hospitals.

The purpose of our study is to investigate the distribution of family physicians in Taiwan and analyze trends over 20 years. Our findings may help to understand the role of physicians and serve as a reference for manpower planning in Taiwan. Furthermore, our study may also be used as a basis for international comparisons of health systems and policy management.

## 2. Materials and Methods

### 2.1. Background

Taiwan is an East Asian island country of 23 million people. In 2018, there were 12,028 medical institutions in Taiwan, excluding institutions for Chinese Medicine and dentistry. These include 25 academic medical centers, 77 metropolitan hospitals, 302 local community hospitals, 44 psychiatric hospitals with new accreditation, 33 unaccredited hospitals, and 11,547 physician clinics [10].

Since 1999, the Joint Commission of Taiwan (JCT) has conducted hospital accreditation and certification throughout Taiwan [11]. Each year, the experts from the JCT review and revise the accreditation requirements for appropriateness and conformance with current policies, laws, and trends in domestic and international health care. Before 2007, the JCT classified medical institutions into four subcategories: academic medical centers, metropolitan hospitals, local community hospitals, and physician clinics. Physician clinics included a small number of institutions that were not yet accredited. The JCT introduced a new system of hospital accreditation in 2007. This caused a discrepancy in data between 2007 and 2010. The total number of institutions remained consistent because the JCT aimed to move away from the previous subcategories, including only hospitals with new accreditation and psychiatric hospitals with new accreditation and physician clinics. The shift took a few years, as medical facilities were due for accreditation at different times. The JCT revised and published the formal accreditation standards in 2010. Hospital institution classifications comprised academic medical centers, metropolitan hospitals, local community hospitals, physician clinics, and psychiatric hospitals with new accreditation. The functions and operations of psychiatric specialty hospitals were different enough to require separate categorization. The classification “hospital with new accreditation” was removed in 2010. The accreditation system implemented in 2010 is still in use today.

In 2018, excluding Chinese Medicine physicians and dentists, there were 47,654 physicians registered in the Taiwan Medical Association (TMA). They comprised 14,071 at academic medical centers, 10,857 at metropolitan hospitals, 4455 at local community hospitals, 481 at psychiatric hospitals with new accreditation, and 17,789 at physician clinics [10].

In Taiwan, medical students can only practice without specialization after completing medical school and passing a two-stage professional examination. Once certified, they join the TMA. The TMA was founded in 1930, and its missions include uniting physicians nationwide, advancing medical skills and common interests, assisting with medical insurance, promoting social services, and collaborating in the development of the medical profession. In 2018, the TMA had 49,000 members [10].

For physicians to qualify for residency, they must first complete post-graduate year (PGY) training. PGY training was introduced in 2003 and has been progressively extended from three months in 2003 to one year in 2011 and two years in 2019. PGY physicians undertake interviews and examinations to qualify for their preferred residency program before the end of their PGY year [11,12].

Family medicine is one of the 21 specialties available, and residents are required to complete three years of hospital-based residency training before sitting for the board examination. Since 1987, except for July 2012 to June 2015 when residency training was 2.5 years, it has been three years [3]. Physicians certified as family medicine specialists in foreign countries and recognized by the Taiwan Association of Family Medicine (TAFM) [13] may also take the board examination. After passing the examination and becoming a family physician, the physician can enroll as a full member of the TAFM. The TAFM was founded in 1986 to promote research and development of family medicine, promote the specialty, raise the standard of primary care, and strengthen ties with international family medicine organizations. There are two types of physicians practicing family medicine: 1. physicians who have passed the family medicine board examination, 2. primary care physicians who applied for certification in family medicine before the board examination was introduced. The latter route was abolished in 1987 [3,14].

In 2018, 513 family physicians worked at academic medical centers, 647 in metropolitan hospitals, 342 in local community hospitals, 10 in psychiatric hospitals with new accreditation, and the remaining 2143 at physician clinics [10].

### 2.2. Data Sources

We obtained the following datasets from the TMA: the statistical yearbooks regarding all practicing physicians and health care organizations in Taiwan. Every yearbook contains the number of members in the association, the distribution of practicing physicians in different cities or counties and practice locations, the age and sex of practicing physicians, the age at death of practicing physicians and their causes of death, and the medical degrees, specializations and original domiciles of practicing physicians. In addition, the number of medical students in training in Taiwan is recorded.

We obtained statistical data of members per year from meeting minutes of TAFM annual general meetings, in which the total number of official members, associate members, and online members is recorded each year. Official members are defined as either (1) an associate member who has passed the family medicine board examination, or (2) a non-TAFM associate member who has passed both the two-stage professional examination and the family medicine board examination; introduced by two TAFM official members and approved by the Board of Directors. Associate members are those who have passed the two-stage professional examination and have been introduced by two official members, approved by the Board of Directors, and reported to the Ministry of Internal Affairs for record. Online members are those who registered for membership on the TAFM website. We also obtained the annual number of residents at each training facility from TAFM.

### 2.3. Study Design

We extracted the total number of physicians practicing in family medicine each year under the “Physician statistics by medical institution” section from the statistical yearbooks from the TMA. We included academic medical centers, metropolitan hospitals, local community hospitals, hospitals with new accreditation, and psychiatric hospitals with new accreditation in the category “hospitals.” All other institutions that provide primary health care for patients were “Physician clinics.” We then analyzed the 20-year trend by comparing the number of family physicians working in hospitals and physician clinics.

In Taiwan, physicians qualified in more than one specialty can only practice one of them when registering with the TMA. Physicians can also opt to practice as general physicians without specialization. We also obtained the total number of physicians in the 21 specialties and general practice under the “Physician statistics by medical institution” section from the statistical yearbooks. Besides family medicine, internal medicine, surgery, gynecology, pediatrics, and general practice, we classified the remaining specialties under “others”. We then analyzed the total number of physicians in each category and compared the 20-year trend.

We used the total number of official members of the TAFM to analyze the increase in family medicine specialists over the past 20 years. We then compared the total number of TAFM official members with the number of TMA members practicing family medicine over the same period.

Next, we obtained the annual number of residents at each training facility from the TAFM and analyzed the 20-year trend. The period when the statistics were collected each year relates to the start of training but varied slightly. Before 2016, it covered each year to June 31, but from 2017 onwards, it covered the year to July 31. For accuracy, we used the actual number of first-year residents enrolled at each hospital as the training capacity of each hospital instead of the “available approved vacancies” of the Ministry of Health and Welfare.

Lastly, we summed up the number of residents in family physician training (first-year to third-year residents). However, complete data are only available from 2002 onwards. We made a correlation in the graph “Distribution of Family Physicians in various practice settings” to analyze the ratio of residents to the total number of family physicians. We subtracted the number of residents in training from those practicing family medicine in hospitals to obtain the percentage of family physicians working at hospitals.

### 2.4. Statistical Analysis

We manually input our collected data and performed our analysis with Microsoft Excel 365 (Microsoft Inc., Redmond, WA, USA). The results were shown in descriptive statistics.

### 2.5. Ethical Approval

The data obtained for this study are all public. In accordance with the Personal Data Protection Act and human research regulations in Taiwan, this study does not require Institutional Review Board (IRB) review.

## 3. Results

### 3.1. Changes in the Ratio of Physicians Registered as Family Physicians in TMA and Number of Members Enrolled in TAFM in Taiwan from 1999 to 2018

The number of TMA physicians practicing in family medicine and the number of TAFM members have increased steadily over the past two decades, with an average annual increase of 89 and 68 physicians, respectively, between 1999 and 2018 (Figure 1). Compared to 48.1% in 1999, the number of registered family physicians in the TMA was 69.5% of the total official members of the TAFM in 2018, which is a smaller difference in the ratio of physicians registered as family physicians in the TMA and number of members enrolled in the TAFM in Taiwan (Appendix A).

### 3.2. Number of Physicians Registered in Different Specialties in TMA in Taiwan from 1999 to 2018

The number of practicing physicians increased yearly nationwide, and increased overall by 19,695 physicians from 1999 to 2018 (Appendix A). The ratio of physicians practicing family medicine to other specialties has increased slightly from 6.7% to 7.7% over the 20 years. It remained stable between 7.6 and 8% from 2004 to 2018 (Figure 2).

### 3.3. Annual Total Training Capacity for Family Medicine from 1999 to 2018

The number of first-year residents enrolled in family medicine nationwide was 129 residents in 1999 and 131 residents in 2018, with an average of 138.9 residents per year from 1999 to 2018. The lowest number was in 2012, with 62 residents, and 2005 the highest with 179 residents. The number of residents in 2011 and 2012 was significantly lower than in other years due to implementing changes in PGY training years. The number stabilized in 2014, and training capacity each year has remained constant with 129.6 residents per year in the last five years (Figure 3).

### 3.4. Distribution of Family Physicians in Various Practice Settings in Taiwan from 1999 to 2018

In 20 years, family physician numbers in academic medical centers nationwide have increased from 284 to 513. In metropolitan hospitals, they increased from 265 to 647, and in local community hospitals, from 212 to 342. Family physicians practicing in psychiatric hospitals with new accreditation increased from 7 to 10 from 2008 to 2018. The number of family physicians practicing in physician clinics also increased from 1115 to 2143 physicians over the past 20 years. The distribution of family physicians in various practice settings in Taiwan from 1999 to 2018 appears unchanged when the 2006–2009 data are excluded (Figure 4).

Overall, the number of family physicians practicing in hospitals was 40.6% of all family physicians practicing in 1999 and increased slightly to 41.4% in 2018. Over the past 20 years, it varied between 37.1% and 42.8%, about two-fifths of total practicing family physicians, and remained stable.

### 3.5. Changes in the Number of Residents in Family Physician Training and Ratio of Family Physicians in Various Practice Settings in Taiwan from 2002 to 2018

The number of residents in family medicine training nationwide compared to total physicians practicing in family medicine decreased from 15.7% in 2002 to 10.4% in 2018 (Figure 5). Excluding residents, the ratio of physicians practicing family medicine in hospitals was about one-third of total practicing family physicians and ranged from 29% in 2002 to 35% in 2018, with an outlier decline to 24% in 2006 (Figure 6).

## 4. Discussion

This study provides a 20-year trend analysis, from 1999 to 2018, of the distribution of family physicians in Taiwan. Our major finding was that family physicians practicing in hospitals accounted for two-fifths of the total practicing family physicians throughout the 20 years studied. Even after subtracting the residents who had not yet qualified but are also registered as family physicians in the TMA, family physicians practicing in hospitals still accounted for one-third of the total. This result was surprisingly high. Family physicians in Western countries, such as the United States and Canada, mainly serve in physician clinics [9,15]. The consistent trend of as many as one-third of family physicians working at hospitals in Taiwan as shown in our study is unique and worth highlighting.

The 2018 Declaration of Astana [16] emphasized the importance of primary health care in the pursuit of global health. The World Organization of Family Doctors (WONCA) also stressed the indispensable role of family physicians in the implementation of the Astana declaration [17,18].

Besides physician clinics, there is a demand for family physicians at every level of medical institutions, even in psychiatric hospitals. This may be because family physicians have trained in comprehensive patient care, which ranges from outpatient to ward care [19,20]. Other than treating acute and chronic illness, family physicians are also responsible for tasks such as health education, health risk assessment, health consultation, and public health planning. Family physicians working at hospitals often have additional responsibilities as compared to those working at physician clinics. For example, in the two academic medical centers in Taiwan’s capital, Taipei Veterans General Hospital [21] and National Taiwan University Hospital [22], the Departments of Family Medicine provide a range of services. In addition to general outpatient services, they also promote cancer screening, smoking cessation, adult health screening, and geriatric health screening in accordance with the National Health Service of the Ministry of Health and Welfare public health policies. They also provide travel medicine clinics in conjunction with the Disease Control Bureau’s “International Travel Preventive Vaccination” service. Family physicians at these hospitals also assist in health administration, general physical examinations for post-marital pre-pregnancy women, pre-nursing home admission, driver’s licenses, expatriate residency applications, and women’s health care. Family physicians at medical centers may also oversee home care and plan and execute public vaccination programs.

The high percentage of up to two-fifths of family physicians at hospital-based practices in Taiwan is partly due to the residents who are also registered as family physicians in the TMA. Family medicine residents are only trained in hospitals. In Taiwan, for example, in 2018, 68 hospitals [23] offered 124 family medicine residency positions, which included metropolitan and local community hospitals and academic medical centers. Family medicine residents in other countries such as Hong Kong [24], the Netherlands [25], and African countries [26] spend only part of their time in hospital training. In Hong Kong, even without formal residency training, they can practice as a family physician if they have the requisite years of training in a primary care facility [24].

As hospitals conduct family medicine residency training, this may indirectly increase the cost of training and requirement for additional faculty staff in the department. Family medicine training focuses on outpatient training, which is different from other departments. Following TAFM board regulations, residents must see patients in the outpatient clinic once a week in their first year of residency. This increases by one session per year. Outpatient clinics require the supervision of a senior resident or attending physician. In addition, during the three years of training, they have three-to-eight-month rotations to primary care institutions (e.g., rural health clinics) [3,27], during which they are absent from the hospital. Besides the need for allocations of teaching staff in proportion to the number of residents, the department also needs to hire more family physicians to share the large workload.

The high proportion of family physicians in hospital-based practice is also probably due to the rich resources and opportunities available at hospitals compared to physician clinics. Hospitals offer the convenience of additional medical diagnostic tests and referrals to other specialists, which aids in diagnosis and follow-up. Some family physicians choose to pursue other subspecialties in the hospital, such as geriatrics or hospice and palliative care [3]. Family physicians training or practicing in subspecialties are still registered under family medicine at the TMA. A 2013 study analyzing inpatient hospice physicians found that family physicians accounted for 37.3% of them, the highest of all specialties [28]. This suggested that 9% of all family physicians practicing in hospitals stayed in practice because they oversaw inpatient hospice and palliative care. Some family physicians may have a passion for teaching, research, or management [29], which is not possible in a clinic setting. Some family physicians stay in hospitals for continuing education credits. Residents may also choose to remain in the same hospital after completing their training out of familiarity.

However, the demand for family physicians at all levels of hospitals does not fully explain why one-third of family physicians in Taiwan are working at hospitals. Our study supports the hypothesis that there is an imbalance between the supply and demand of family physicians in Taiwan.

First, our study showed a discrepancy between the number of physicians registered under family medicine in the TMA and the number of members of the TAFM over 20 years. The number of practicing family physicians in the TMA was continuously lower than the total number of TAFM official members, which indicated about 30.5% of family physicians were certified but not practicing in 2018. This discrepancy may be because the number of TMA members represents residents under family medicine training and certified family physicians, whereas TAFM members include a larger population, such as family physicians in practice, those not practicing, or practicing in other specialties. Before the residency program and board examination were introduced in 1987 [14], physicians could become family physicians solely by passing the board examination. Hence, many of the family physicians during that period may be dual-boarded. However, dual-boarded physicians are only allowed to register in one of their specialties in the TMA [30]. They may be an official member of the TAFM but registered as another specialty instead of family medicine in the TMA. In other countries, such as the United States, dual-boarded pediatrics and internal medicine physicians can practice in both specialties concurrently [31].

However, this gap is narrowing year by year, with a greater proportion of certified family physicians in practice. There may be several reasons. In Taiwan, physicians must undergo residency training before taking the specialty exams. Medical students are around 24 years old when they graduate after 6 to 7 years of training. Training for a second or third specialty will be time-consuming and is rare [32]. As a result, the number of dual-boarded physicians has decreased dramatically in recent years. Furthermore, some physicians who qualified in family medicine have either retired or passed away. Despite the narrowing gap, the absolute number of family physicians is insufficient to compensate for the members dropping out of the association. Hence, we deduce that the rate of increase in practicing family physicians is still not enough to meet the demand.

Our study suggests that the total supply of family physicians is low by reviewing registered specialties in the TMA in Taiwan and the total training capacity for family medicine from 1999 to 2018. The annual capacity for family medicine residency has not changed much in the last 20 years; in 2018, it was 7.6% of Taiwan’s total specialist training capacity [33]. In the United States, it is 12.6% [34]. Even so, it is not enough, and the American Academy of Family Physicians (AAFP) has proposed a goal of 25 × 2030 [35] to meet their national primary care manpower needs. One out of every four physicians will be enrolled in family medicine residency training by 2030.

The ratio of physicians practicing family medicine in the TMA compared to other specialties has also not grown significantly over our study period. Compared with other specialties, family physicians only account for an average of 7.7% of all physicians. This percentage is generally lower than in countries such as the United States or members of the European Union [35,36], where family physicians account for a higher proportion, about 12% of total physicians. In Spain, family physicians accounted for 38.1% of all physicians in 2009 [37]. Moreover, not all physicians who specialize in family medicine will practice family medicine; some physicians choose general practice [38]. Hence, training capacity should be increased to meet the demand for family physicians.

Overall, our study shows that, unlike in other countries, family physicians in hospital practice consistently account for about one-third of all family physicians in practice in Taiwan. Although all levels of hospitals need a steady number of family physicians to support the large workload and public health responsibilities, this proportion would drop if more family physicians worked in physician clinics. The number of practicing family physicians and residents trained annually appears insufficient to meet the demand for family physicians. Thus, without an increase in the denominator, there will likely be no extra manpower to increase family physicians conducting primary care at physician clinics. Without policy changes such as delegating some of the public health tasks of hospitals to physician clinics, increasing the number of family physicians trained each year, and reviewing the need for hospital-based training, this problem will persist. The growth of family physicians will not meet current and future demand.

The demand for family physicians in Taiwan will only continue to rise in the upcoming years. According to the National Development Council in Taiwan, Taiwan became an aged society in 2018 and is predicted to enter a “super-aged society” within a short span of eight years. Male and female life expectancy in Taiwan has increased from 73.3 to 77.7 and 79.0 to 84.2, respectively, from 1999 to 2019. The elderly population in Taiwan is expected to continue growing as the baby boomers born between 1946 and 1964 grow older [39]. In Taiwan, 81% of elderly people over the age of 65 have at least one chronic disease [40]. Hence, the demand for family physicians who specialize in treating chronic illnesses will increase. As shown in our study, family medicine residents increased by an average of 129.6 residents annually in the last five years. Therefore, the number of family specialists will only increase by an estimation of 648 physicians five years later. However, the number of elderly people aged 65 and above is projected to increase by 909,180 (4%) from 2021 to 2026 [41]. There is an urgent need for a sustainable supply of family physicians to meet the surge in numbers of patients as Taiwan progresses into a “super-aged society”. This rising demand–supply mismatch of family physicians in Taiwan requires further investigations.

### Limitations

Our study has some limitations. Our study is limited to the period 1999–2018, as older statistics are not available electronically or publicly and the most recent statistics have not yet been published. The statistics we have obtained from the TMA and TAFM are based on a point of time each year. However, the data are not entirely accurate due to changes throughout the year. The number of practicing family physicians in TMA may be overestimated as training residents are also registered under family medicine; however, the exact number of residents without specialty is not available. While we consider first to third-year family medicine residents as residents in training, some year three residents of a 2-year training program may have obtained a specialty in family medicine during the transition period. In addition, some hospitals are included in the physician clinic category because they have not yet been evaluated, resulting in an overestimation of the number of primary care providers. Moreover, because most studies have examined family physicians practicing in urban and rural areas, and fewer have investigated different health care settings, it is difficult to make comparisons across countries. Finally, the roles of family physicians in various health care facilities, which are unknown in this study, warrant further studies.

## 5. Conclusions

Our 20-year trend analysis of family physicians’ worksites in Taiwan showed a high proportion, about one-third, of family physicians in hospital-based practices. This is likely due to a low supply of family physicians. This issue requires urgent attention to maintain national health care quality and prevent a shortage of family physicians in the long run. 

## Figures and Tables

**Figure 1 ijerph-18-09097-f001:**
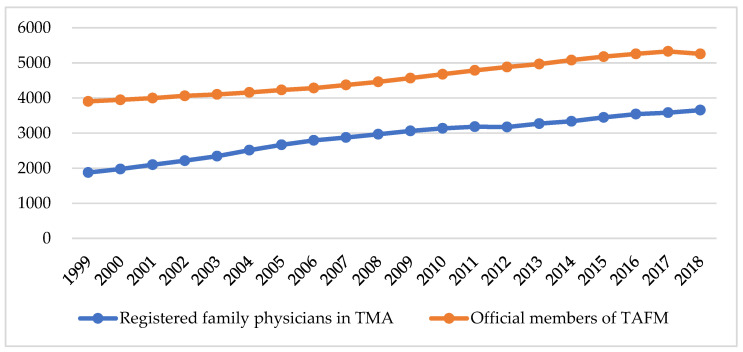
Changes in the ratio of physicians registered as family physicians in TMA and number of members enrolled in TAFM in Taiwan from 1999 to 2018.

**Figure 2 ijerph-18-09097-f002:**
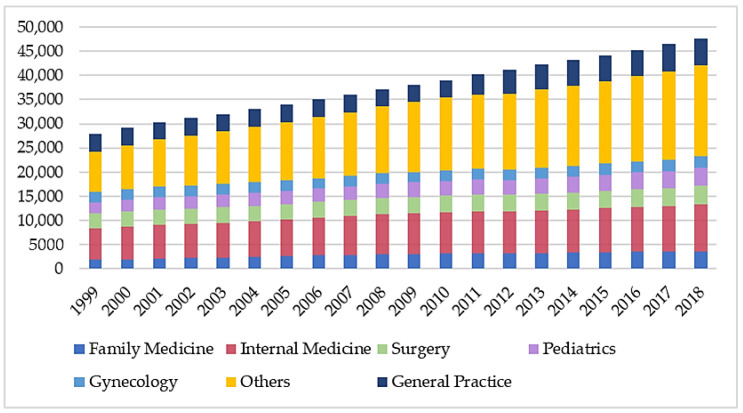
Number of physicians registered in different specialties in TMA in Taiwan from 1999 to 2018.

**Figure 3 ijerph-18-09097-f003:**
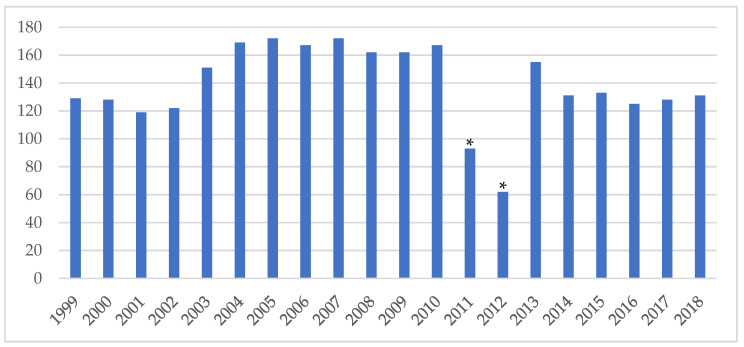
Annual total training capacity for family medicine from 1999 to 2018. * 2011 and 2012 were years of a transition of changes in PGY training years. Training capacity was adjusted according to the number of eligible PGY in that year.

**Figure 4 ijerph-18-09097-f004:**
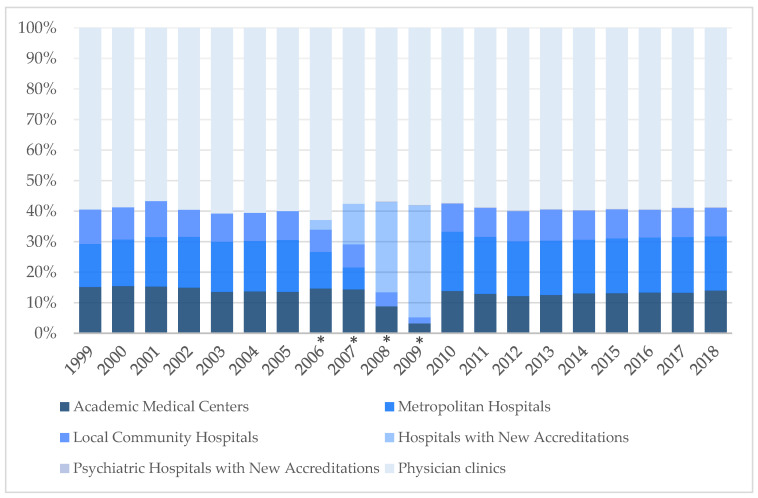
Distribution of family physicians in various practice settings in Taiwan from 1999 to 2018. * Marked transitions in the accreditation regulations in 2006–2009.

**Figure 5 ijerph-18-09097-f005:**
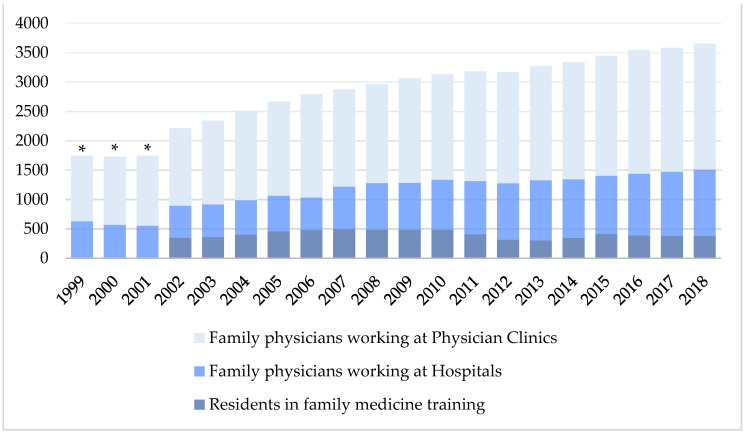
Changes in the numbers of residents in family physician training and number of family physicians in various practice settings in Taiwan from 2002 to 2018. * 1999–2001 data of the number of residents in family medicine training are not available.

**Figure 6 ijerph-18-09097-f006:**
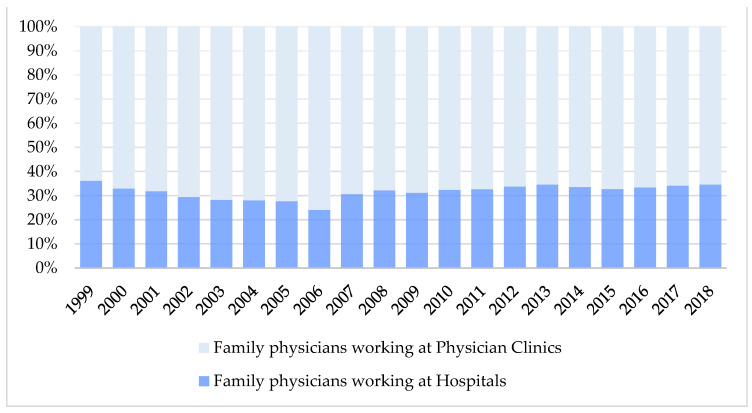
Changes in the ratio of family physicians in various practice settings in Taiwan from 2002 to 2018.

## Data Availability

Data are contained within the article and Appendix A.

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
