# Peer review of "Family Physicians Working at Hospitals: A 20-Year Nationwide Trend Analysis in Taiwan"

_ijerph, 2021, doi:10.3390/ijerph18179097_

Round 1
Reviewer 1 Report
This manuscript presents data on the number and practice location of family physicians in Taiwan between 1999 and 2018. Overall the paper is well-written and the results are clearly presented, and the findings highlight a potential shortage of workforce for family medicine in Taiwan. I only have some minor queries and suggested changes (mainly related to wording), which the authors may wisht to consider:
Page 2, second paragraph: The new system of hospital accreditation introduced in 2007 seems to have a major impact on the data and workforce training. Perhaps some more description of the changes compared with the old system (relevant to data used in this study) could be included in the manuscript? In Figure 4 on page 6, why are ‘Hospitals with New Accreditations’ only shown between 2006 and 2009 and not in subsequent years?
Page 3, line 115: would it be more accurate to say: All ‘other’ institutions that provide primary health care for patients were “physician clinics.”
Page 3, lines 126-137: the description of membership for Taiwan Association of Family Medicine (TAFM) needs some clarification:
- Based on information published in the TAFM website, official membership could be achieved through two different routes: one is “a TAFM associate member who has passed the family medicine board examination” as described, and the other is someone (who is not a TAFM associate member) who has passed both the two-stage professional examination and the family medicine board examination, has been introduced by two TAFM official members and approved by the Board of Directors?
- Are online members a separate (mutual exclusive) category from official members and associate members? If so, what are the requirements to become an online member (apart from registering for membership online)?
Page 4, line 163: it might be clearer to say “….have increased steadily over the past two decades, with an average annual increase of 89…”
Page 4, lines 165-166: why is an increase from 48.1% to 69.5% a ‘smaller difference’?
Page 4, lines 172-173: overall ‘increased’ by 19,695…
Page 4, lines 173-175: based on data presented in Table S2, the description that “The ratio of physicians practicing family medicine to other specialties has steadily increased from 6.7% to 7.7% over the 20 years” is not entirely accurate, as the percentage peaked at 8.1% in 2009 and then dropped slightly in subsequent years. Given that a 1% increase over 20 years is fairly minor, it may be easier to say “…has increased slightly from 6.7% to 7.7% over the 20 years” instead.
Page 5, lines 180-182: the sentences might read better if it starts with “The number of first year residents…” and ends with “with an average of 138.9 residents per year between 1999 and 2018.”
Page 7, Figure 5: is the “Total residents in family medicine training” shown in the figure (i.e. the black crosses) percentages or absolute numbers? If they are percentages, they don’t seem to match with the percentages quoted in the main text? (i.e. line 211 stated “20.2% in 2002 to 13.4% in 2018”, but all black crosses in Figure 5 fall below 20%)
Page 7, Figure 5 & page 6 lines 212-214: although Figure 5 was quoted at the end of the statement “number of physicians practicing family medicine in hospitals was about one-third of total practicing family physicians and ranged from 29% in 2002 to 35% in 2018, with an outlier decline to 24% in 2006”, these percentages cannot be seen easily in Figure 5. Rather than showing percentages, would it be better to show absolute numbers of (1) family medicine residents in hospitals; (2) non-resident family medicine physicians practicing in hospitals; (3) family medicine physicians practicing in physician clinics in a stacked bar chart (like Figure 2)?
Page 8, line 291: continuously ‘lower’ than…
Page 8, lines 293-295: This discrepancy may be because the number of TMA members represents residents under family medicine training and certified family physicians, ‘whereas’ TAFM members include a larger population…
Page 10, line 377: … and ‘does not require institutional board review’.
Author Response
(Please see the attached file for response to all reviewers)
Point 1: Page 2, second paragraph: The new system of hospital accreditation introduced in 2007 seems to have a major impact on the data and workforce training. Perhaps some more description of the changes compared with the old system (relevant to data used in this study) could be included in the manuscript? In Figure 4 on page 6, why are ‘Hospitals with New Accreditations’ only shown between 2006 and 2009 and not in subsequent years?
Response 1: Thank you for your suggestion. Indeed there was a huge transition in the system of hospital accreditation from 2007 to 2011. The main difference in the system is way the Joint Commission of Taiwan (JCT) classified medical institutions. The total number of institutions remained the same. In 2007, the new accreditation system took place, with the goal of moving away from old subcategories (academic medical centers, metropolitan hospitals, local community hospitals). However, after multiple revisions, JCT decided to revert to older subcategories in 2010, abolishing “Hospitals with New Accreditations.” “Psychiatric hospitals with new accreditation” was kept as psychiatric hospitals required exclusive accreditation standards different from general hospitals. We revised this paragraph for better clarity as advised:
“JCT aimed to move away from the previous subcategories, including only hospitals with new accreditation and psychiatric hospitals with new accreditation and physician clinics. The shift took a few years, as medical facilities were due for accreditation at different times. JCT revised and published the formal accreditation standards in 2010. Hospital institution classifications comprised academic medical centers, metropolitan hospitals, local community hospitals, physician clinics, and psychiatric hospitals with new accreditation. The functions and operations of psychiatric specialty hospitals were different enough to require separate categorization. The classification “hospital with new accreditation” was removed in 2010. The accreditation system implemented in 2010 is still in use today.”
Point 2. Page 3, line 115: would it be more accurate to say: All ‘other’ institutions that provide primary health care for patients were “physician clinics.”
Response 2: Thank you. We have edited that line for improved accuracy.
Point 3. Page 3, lines 126-137: the description of membership for Taiwan Association of Family Medicine (TAFM) needs some clarification:
- Based on information published in the TAFM website, official membership could be achieved through two different routes: one is “a TAFM associate member who has passed the family medicine board examination” as described, and the other is someone (who is not a TAFM associate member) who has passed both the two-stage professional examination and the family medicine board examination, has been introduced by two TAFM official members and approved by the Board of Directors?
- Are online members a separate (mutual exclusive) category from official members and associate members? If so, what are the requirements to become an online member (apart from registering for membership online)?
Response 3: Thank you for your detailed review. Online members was defined as members that was registered online, and probably overlapped with official and associate members. However, precise definition was not available in meeting minutes of TAFM annual general meetings. As for the definition of official members, it was an error on our part and we have made correction as follows:
“Official members are defined as either (1) an associate member who has passed the family medicine board examination, or (2) a non-TAFM associate member who has passed both the two-stage professional examination and the family medicine board examination; introduced by two TAFM official members and approved by the Board of Directors.”
Point 4. Page 4, line 163: it might be clearer to say “….have increased steadily over the past two decades, with an average annual increase of 89…”
Response 4: Thank you. We have revised our sentence according to your suggestion.
Point 5. Page 4, lines 165-166: why is an increase from 48.1% to 69.5% a ‘smaller difference’?
Response 5: Thank you for your suggestion. By a “smaller difference”, we meant a smaller gap in the number of family physicians certified and practicing (which is represented by number of TAFM members and TMA family physicians respectively). We apologize for the confusion, and we have modified our statement as follows:
“Compared to 48.1% in 1999, the number of registered family physicians in TMA was 69.5% of the total official members of TAFM in 2018, which is a smaller difference in the ratio of physicians registered as family physicians in TMA and number of members enrolled in TAFM in Taiwan.”
Point 6. Page 4, lines 172-173: overall ‘increased’ by 19,695…
Response 6: We have made the amendment as advised.
Point 7. Page 4, lines 173-175: based on data presented in Table S2, the description that “The ratio of physicians practicing family medicine to other specialties has steadily increased from 6.7% to 7.7% over the 20 years” is not entirely accurate, as the percentage peaked at 8.1% in 2009 and then dropped slightly in subsequent years. Given that a 1% increase over 20 years is fairly minor, it may be easier to say “…has increased slightly from 6.7% to 7.7% over the 20 years” instead.
Response 7: Thank you for your advice, we have corrected our statement.
Point 8. Page 5, lines 180-182: the sentences might read better if it starts with “The number of first year residents…” and ends with “with an average of 138.9 residents per year between 1999 and 2018.”
Response 8: Thank you for your suggestion. We have made the changes.
Point 9. Page 7, Figure 5: is the “Total residents in family medicine training” shown in the figure (i.e. the black crosses) percentages or absolute numbers? If they are percentages, they don’t seem to match with the percentages quoted in the main text? (i.e. line 211 stated “20.2% in 2002 to 13.4% in 2018”, but all black crosses in Figure 5 fall below 20%)
Response 9: Thank you so much for pointing out this error. The graph is shown correctly but writing data was incorrectly entered. We have revised our figure description as follows:
“The number of residents in family medicine training nationwide compared to total physicians practicing in family medicine decreased from 15.7% in 2002 to 10.4% in 2018.”
Point 10. Page 7, Figure 5 & page 6 lines 212-214: although Figure 5 was quoted at the end of the statement “number of physicians practicing family medicine in hospitals was about one-third of total practicing family physicians and ranged from 29% in 2002 to 35% in 2018, with an outlier decline to 24% in 2006”, these percentages cannot be seen easily in Figure 5. Rather than showing percentages, would it be better to show absolute numbers of (1) family medicine residents in hospitals; (2) non-resident family medicine physicians practicing in hospitals; (3) family medicine physicians practicing in physician clinics in a stacked bar chart (like Figure 2)?
Response 10: Thank you for your suggestion. We agree that using absolute numbers would be a better way of presenting our data, but we also hope to present out data using percentages to emphasize the ratio of one-third for better comparison with other countries in our discussion. Hence after much consideration, we decided to modify Figure 5 as suggested and present the ratio of family physicians working at hospitals and physician clinics in a separate Figure 6. The modification to this part of our results is as follows:
“3.5. Changes in the number of residents in family physician training and ratio of family physicians in various practice settings in Taiwan from 2002 to 2018
The number of residents in family medicine training nationwide compared to total physicians practicing in family medicine decreased from 15.7% in 2002 to 10.4% in 2018 (Fig. 5). Excluding residents, the ratio of physicians practicing family medicine in hospitals was about one-third of total practicing family physicians and ranged from 29% in 2002 to 35% in 2018, with an outlier decline to 24% in 2006 (Fig. 6).
[Please see the attachment for the updated figures]
Point 11. Page 8, line 291: continuously ‘lower’ than…
Response 11: We have made the amendment as advised.
Point 12. Page 8, lines 293-295: This discrepancy may be because the number of TMA members represents residents under family medicine training and certified family physicians, ‘whereas’ TAFM members include a larger population…
Response 12: Thank you for your suggestion, we have added the conjunction for better expression.
Point 13. Page 10, line 377: … and ‘does not require institutional board review’.
Response 13: Thank you for your kind suggestion, we have made the corresponding changes.

Reviewer 2 Report
Thank you for giving me to review your manuscript. This manuscript is interesting and meaningful for considering the working conditions of family physicians. Regarding the contents, the following revision should be considered for the quality of research.
- The title may not show a clear image of research contents. Based on the results, not just one-third. The author should describe the title, including research designs.
- The background is too short, and there are no theoretical frameworks regarding family physicians' deployment in countries and clear research questions.
- In the introduction, there are few references to support this research topic. The authors should do more regarding this research topic.
- The introduction should clearly include the research question and rationale of this study, including the advantage of this study.
- The structure of the manuscript is flawed. This manuscript is difficult to read for international readers. The authors should write the correct contents in correct locations in the form of original research.
- In the sample section of the method, there are no descriptions regarding sample calculation. Therefore, the authors should descript the sample size calculation.
- This is quantitative research. This research should contain statistical analysis for validity and reliability.
- The discussion should describe the limitation of sampling bias and the results' applicability to other settings, and the future investigation in the limitation part.
Author Response
(Please see the attached file for response to all reviewers)
Point 1: The title may not show a clear image of research contents. Based on the results, not just one-third. The author should describe the title, including research designs.
Response 1: Thank you for your suggestion. Figure 5 of our results showed that the actual number of physicians practicing family medicine in hospitals was about one-third of total practicing family physicians. This is when we excluded the number of residents who had not yet qualified but are also registered as family physicians. However, to avoid confusion and improve the precision of our title, we have changed our title to “Family Physicians Working at Hospitals: A 20-year Nationwide Trend Analysis in Taiwan.”
Point 2: The background is too short, and there are no theoretical frameworks regarding family physicians' deployment in countries and clear research questions.
Response 2: Thank you for the detailed review. Indeed, the introduction should present the background to the study in a concise manner to provide readers with the essential context needed to understand the research problem. In the introduction, we provided a general overview of the job scope of family physicians and preference of practice locations of most family physicians in most western countries. Indeed, detailed background is important and crucial before moving on to discuss about the working locations of family physicians in Taiwan. Hence after giving a general background in the introduction, we presented more background details regarding the family physicians’ system in Taiwan under “2.1 Background” of the Method section.
The family systems vary largely across different countries, and it is difficult to find a theoretical framework that fits all. Our study is a descriptive analysis study of the practice locations of family physicians in Taiwan, and no experiments were conducted. The value of this study is in that we pointed out the unique phenomenon of family physicians working at hospitals instead of clinics, which is uncommon in other non-authoritarian countries. When freely deployed, most family physicians in these countries tend to choose to work at clinics as primary care physicians. Reasons behind this phenomenon certainly warrants further studies.
Point 3: In the introduction, there are few references to support this research topic. The authors should do more regarding this research topic.
Response 3: Thank you for the suggestion. Finding references to support this research topic was rather difficult due to limited articles related to this topic. There are many studies discussing the geographical distribution of family physicians but rarely practice locations. As mentioned in point 2, most family physicians in other non-authoritarian countries choose to work at primary care clinics when they are freely deployed. There are exceptions to this, such as family physicians working at rural areas of Canada, as mentioned in our introduction “A minority of family physicians work at community or remote hospitals, where family physicians may be among the few physicians employed in these sparsely populated areas.” However, there have been an exodus from hospitals to clinics in Canada as well, as presented in our introduction. The consistent trend of as many as one-third family physicians working at hospitals in Taiwan as shown in our study is different from other countries and worth highlighting.
We have expanded our introduction according to the reviewer’s suggestion. The additional text is as follows:
“Most studies in the past focused on geographic maldistribution of family physicians, and few on the distribution of family physicians between private practices and hospitals.”
Point 4: The introduction should clearly include the research question and rationale of this study, including the advantage of this study.
Response 4: Thank you for your kind suggestion. Our study is a descriptive analysis study, and no experiments were conducted. We hope to present the distinctive finding of family physicians working at hospitals in Taiwan, a finding that is uncommon in other non-authoritarian countries. Hence, there is no research question involved. To improve clarity, we have revised the paragraph as follows:
“The purpose of our study is to investigate the distribution of family physicians in Taiwan and analyze trends over 20 years. Our findings may help to understand the role of physicians and serve as a reference for manpower planning in Taiwan. Furthermore, our study may also be used as a basis for international comparisons of health systems and policy management.”
To further stress the importance and uniqueness of this study, we also added the following text to our discussion:
“The consistent trend of as many as one-third family physicians working at hospitals in Taiwan as shown in our study is unique and worth highlighting.”
Point 5: The structure of the manuscript is flawed. This manuscript is difficult to read for international readers. The authors should write the correct contents in correct locations in the form of original research.
Response 5: We have organized our manuscript according to the latest IJERPH template and according to the international scientific IMRaD format. Thank you for the suggestion, we have subcategorized the Methods section which was previously too lengthy for easier reading.
2.1. Background
2.2. Data sources
2.3. Study design
2.4. Statistical analysis
2.5. Ethical approval
Point 6: In the sample section of the method, there are no descriptions regarding sample calculation. Therefore, the authors should descript the sample size calculation.
Response 6: Sorry for any misunderstanding. Our data collected were complete secondary dataset without sampling from the Taiwan Medical Association and Taiwan Association of Family Medicine. Thus, no sample size calculation was involved.
Under section 2.2 data source of the Method section:
“TMA: the statistical yearbooks regarding all practicing physicians and health care organizations in Taiwan.”
“Statistical data …of TAFM…, in which the total number of official members…is recorded each year.”
Point 7: This is quantitative research. This research should contain statistical analysis for validity and reliability.
Response 7: Thank you for your advice. Our study was presented in descriptive statistics, hence there was no statistical analysis for validity and reliability. However, we revised section 2.4 statistical analysis of the Method section for better precision:
“We manually input our collected data and performed our analysis with Microsoft Excel 365 (Microsoft Inc., Redmond, WA, USA). The results were shown in descriptive statistics.”
Point 8: The discussion should describe the limitation of sampling bias and the results' applicability to other settings, and the future investigation in the limitation part.
Response 8: Thank you for the thorough review. As mentioned in point 6, the dataset we obtained were data of the whole target population, so sampling was not done. Our results, although interesting, may not be applicable to other settings as the phenomenon of a significant proportion of family physicians choosing to work at hospitals is truly rare and possibly only present in Taiwan.
We have made addition as follows to the Limitations section as recommended by the reviewer:
“Finally, the roles of family physicians in various healthcare facilities, which are unknown in this study, warrants further studies.”
Point 9: Moderate English changes required.
Response 9: Thank you for your recommendation. This article has been reviewed by a professional English editorial company, and the revision will be corrected again for typographical errors, grammar, and phrases as suggested.

Reviewer 3 Report
The paper is very interesting and analyze the trends in practice locations of family physicians in Taiwan between 1999 and 2018. However, the paper just presents a general descriptive analysis of the Taiwan Association of Family Medicine and Taiwan Medical Association.
I suggest the follow improvements:
- Consider a more recent dataset to 2019 or 2020.
- Standardize the colors on presented graphs. Figure 4 does not have a good choice around color pallete.
- Prepare a prediction section relative the future demands of Family Physicians in Taiwan for the next few years (5 years ahead) and add a detailed discussion about this prediction.
Author Response
(Please see the attached file for response to all reviewers)
Point 1: Consider a more recent dataset to 2019 or 2020.
Response 1: Thank you for your suggestion. The 2020 statistical yearbook has not yet been published, possibly due to the COVID pandemic. As mentioned under the Limitation section: “Our study is limited to the period 1999–2018, as older statistics are not available electronically or publicly, and the most recent statistics have not yet been published.” The 2019 yearbook was published after we prepared our manuscript. As our study focused on a 20-year trend, we did not include the 2019 statistics. We have included an updated Figure 4 graphs with the 2019 data included for your interest. The data for 2019 was consistent with the trend of our study.
[Please see the attachment for the figure with 2019 data]
Point 2: Standardize the colors on presented graphs. Figure 4 does not have a good choice around color palette.
Response 2: We have standardized the colors on presented graphs according to the reviewer’s suggestion.
[Please see the attachment for updated color scheme]
Point 3: Prepare a prediction section relative the future demands of Family Physicians in Taiwan for the next few years (5 years ahead) and add a detailed discussion about this prediction.
Response 3: Thank you for your suggestion. We have added a paragraph to our discussion as follows:
“The demand for family physicians in Taiwan will only continue to rise in the upcoming years. According to the National Development Council in Taiwan, Taiwan became an aged society in 2018 and is predicted to enter a “super-aged society” within a short span of eight years. Male and female life expectancy in Taiwan has increased from 73.3 to 77.7 and 79.0 to 84.2 respectively from 1999 to 2019. The elderly population in Taiwan is expected to continue growing as the baby boomers born between 1946 and 1964 grow older [39]. In Taiwan, elderly people with at least one chronic disease over the age of 65 is about 81% [40]. Hence, the demand for family physicians, who specializes in treating chronic illnesses, will increase. As shown in our study, family medicine residents increased by an average of 129.6 residents annually in the last five years. Therefore, the number of family specialists will only increase by an estimation of 648 physicians five years later. However, the number of elderly people aged 65 and above is projected to increase by 909,180 (4%) from 2021 to 2026[41]. There is an urgent need for a sustainable supply of family physicians to meet the surge in numbers of patients as Taiwan progresses into a “super-aged society”. This rising demand-supply mismatch of family physicians in Taiwan requires further investigations.”
[39] Population projections for the Republic of China (Taiwan): 2020-2070, Department of Human Resources Development and National Development Council. Available online: https://pop-proj.ndc.gov.tw/upload/download/Population%20Projections%20for%20the%20Republic%20of%20China%20(Taiwan)-2020~2070.pdf (accessed on 14 August 2021).
[40] Chou, C.Y., Chiu, C.J., Chang, C.M., Wu, C.H., Lu, F.H., Wu, J.S., & Yang, Y. C. Disease-related disability burden: a comparison of seven chronic conditions in middle-aged and older adults. BMC geriatr. 2021, 21, 1-11, doi: 10.1186/s12877-021-02137-6.
[41] Population by broad age groups: National Development Council. Available online: https://pop-proj.ndc.gov.tw/main_en/dataSearch6.aspx?uid=78&pid=78 (accessed on 14 August 2021).

Round 2
Reviewer 2 Report
The manuscript has been considerably improved. I think that this paper is suited for inclusion in our journal.
Author Response
Thank you for your review. We are very grateful for your advices!